# Caerulomycin and collismycin antibiotics share a *trans*-acting flavoprotein-dependent assembly line for 2,2'-bipyridine formation

Bo Pang [1,4], Rijing Liao [2,4], Zhijun Tang [1,4], Shengjie Guo[1], Zhuhua Wu[1] & Wen Liu [1,3✉]

Linear nonribosomal peptide synthetases (NRPSs) and polyketide synthases (PKSs) template the modular biosynthesis of numerous nonribosomal peptides, polyketides and their hybrids through assembly line chemistry. This chemistry can be complex and highly varied, and thus challenges our understanding in NRPS and PKS-programmed, diverse biosynthetic processes using amino acid and carboxylate building blocks. Here, we report that caerulomycin and collismycin peptide-polyketide hybrid antibiotics share an assembly line that involves unusual NRPS activity to engage a *trans*-acting flavoprotein in C-C bond formation and hetero-cyclization during 2,2'-bipyridine formation. Simultaneously, this assembly line provides dethiolated and thiolated 2,2'-bipyridine intermediates through differential treatment of the sulfhydryl group arising from ʟ-cysteine incorporation. Subsequent ʟ-leucine extension, which does not contribute any atoms to either caerulomycins or collismycins, plays a key role in sulfur fate determination by selectively advancing one of the two 2,2'-bipyridine intermediates down a path to the final products with or without sulfur decoration. These findings further the appreciation of assembly line chemistry and will facilitate the development of related molecules using synthetic biology approaches.

---

[1] State Key Laboratory of Bioorganic and Natural Products Chemistry, Center for Excellence in Molecular Synthesis, Shanghai Institute of Organic Chemistry, University of Chinese Academy of Sciences, Shanghai, China. [2] Shanghai Institute of Precision Medicine, Ninth People's Hospital, Shanghai Jiao Tong University School of Medicine, Shanghai, China. [3] Huzhou Center of Bio-Synthetic Innovation, Huzhou, China. [4]These authors contributed equally: Bo Pang, Rijing Liao, Zhijun Tang. ✉email: wliu@sioc.ac.cn

Linear nonribosomal peptide synthetases (NRPSs) and polyketide synthases (PKSs) are often large molecular machines that are composed of multidomain modules[1–6]. They have evolved through functional unit (e.g., protein, module, or domain) combination and variation to afford various assembly lines that program diverse polymerization and modification processes using amino acids, short carboxylic acids, or both as monomers. Assembly line chemistry, which can be complex and highly varied, challenges our understanding of how nature creates numerous nonribosomal peptides, polyketides, and hybrids thereof that historically play critical roles in medicinal chemistry and chemical biology. The biosynthesis of 2,2'-bipyridine, the core structure of a large class of synthetic bidentate metal-chelating ligands with a variety of applications in many areas of chemistry[7,8], occurs via such a process.

2,2'-Bipyridine natural products include caerulomycins (e.g., CAE-A, Fig. 1), a group of structurally related antibiotics that possess potent immunosuppressive activity[9–13]. Although the process through which 2,2'-bipyridine is formed remains poorly understood, we and others elucidated first details of the biogenesis of CAEs and identified an NRPS-PKS assembly line that contains three modular proteins, i.e., CaeA1, CaeA2, and CaeA3, to template a peptide-polyketide hybrid skeleton for 2,2'-bipyridine formation (Fig. 1)[14–16]. CaeA1 is a bifunctional protein composed of a peptidyl carrier protein (PCP) and an adenylation

(A) domain. It was proposed to incorporate picolinic acid, which arises from the activities of lysine aminotransferase CaeP1 and subsequent oxidase CaeP2,[14,16] as the starter unit for molecular assembly and thus provide the unmodified pyridine unit (Ring **B**) of CAEs. CaeA2 is a hybrid protein, with a typical PKS module containing a ketosynthase (KS), acyltransferase (AT), and acyl carrier protein (ACP) and an atypical NRPS module containing a condensation/cyclization (Cy) domain, A domain, PCP, and terminal C (Ct) domain. Logically, this protein can incorporate malonyl-CoA and l-cysteine in tandem and associate CaeA1 to provide the essential building blocks for constructing the di- or trisubstituted pyridine unit (Ring **A**) of CAEs[14,17,18]. As proposed, the PKS module of CaeA2 catalyzes two-carbon elongation and connects the atoms C3 and C4 to C2, the carboxyl carbon of the starter picolinyl unit. The atypical NRPS then catalyzes l-cysteine extension, which is unique and occurs at Cβ of the l-cysteine residue to form a C–C bond, instead of its α-amino group for amide bond formation as usual, and thus provides C5, C6, and N1 as well as exocyclic C7 after heterocyclization. To our knowledge, peptidyl elongation through a C–C linkage lacks precedent in NRPS catalysis. The domains of CaeA3 are organized as C-A-PCP for l-leucine extension[14,15]. Despite not contributing any atoms in the mature products, CaeA3 activity is necessary for the formation of 2,2'-bipyridinyl-l-leucine (**1**), an offline intermediate that can be specialized to individual CAE

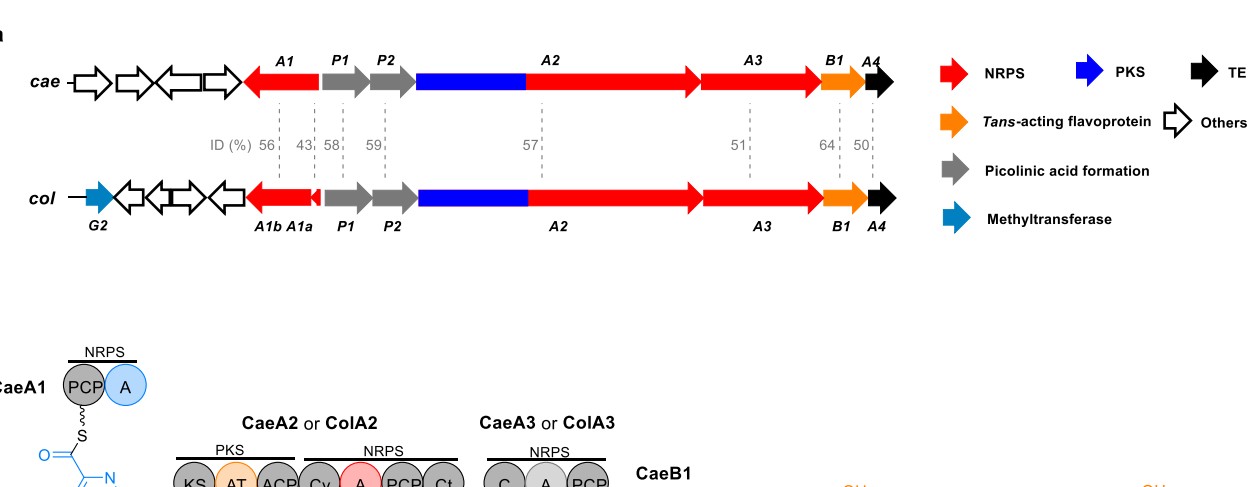

**Fig. 1 Biogenesis of the polyketide-peptide hybrid, 2,2'-bipyridine antibiotics CAE-A and COL-A. a** Organization of related genes in the CAE and COL biosynthetic gene clusters that codes for the hybrid NRPS/PKS assembly lines, the associated *trans*-acting flavoproteins and other proteins. The didomain encoding gene *caeA1* in the *cae* cluster corresponds to the two discrete genes, *colA1a* and *colA1b*, in the *col* cluster. Gene functions are annotated by color on the left. ID, sequence identity. **b** Hybrid NRPS/PKS assembly line for 2,2'-bipyridine formation. PKS and NRPS modules are indicated. The functional domains (i.e., AT and A domains) for substrate specificity are indicated in different colors with their associated building blocks: blue for picolinyl, yellow for malonyl, red for l-cysteinyl, and light gray for l-leucinyl.

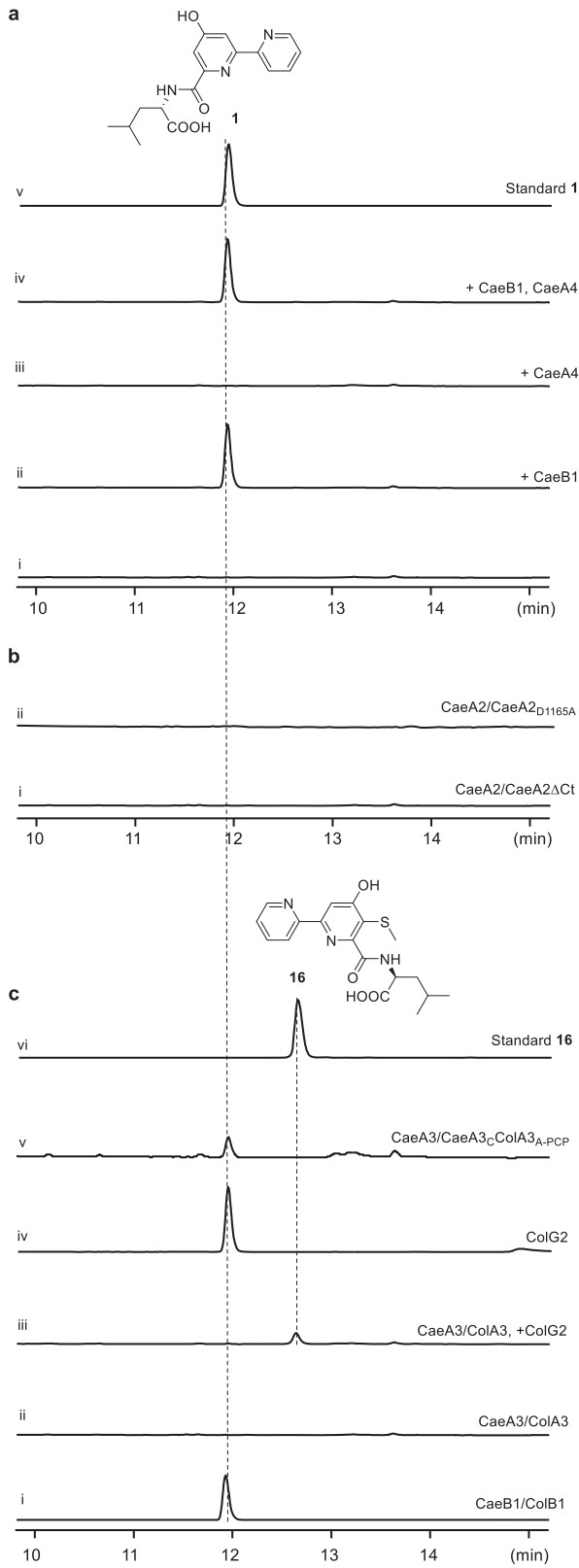

**Fig. 2 In vitro reconstitution of the 2,2′-bipyridine assembly line.** 2,2′-Bipyridine products were examined by HPLC (λ 315 nm). Each assay was performed at least three times, and every time at least two parallel samples were used. **a** Determination of the *trans*-acting partner. In the reaction mixture where the proteins CaeA1, CaeA2, and CaeA3 were combined with the substrates picolinic acid, malonyl-CoA, L-cysteine, and L-leucine as well as ATP (i), CaeB1 (ii), CaeA4 (iii), or both CaeB1, and CaeA4 (iv) were added individually. Synthetic **1** was used as a standard (v). **b** Validation of the necessity of the Ct and Cy domains in the atypical NRPS module for 2,2′-bipyridine formation. In the above CaeB1-containing, **1** producing reaction mixture, wild-type CaeA2 was replaced with truncated CaeA2ΔCt (i) and mutated CaeA2$_{D1165A}$ (ii), respectively. **c** Determination of the similarity and difference in the formation of dethiolated and thiolated 2,2′-bipyridines by protein/domain exchange. In the above CaeB1-containing, **1** producing reaction mixture, CaeB1 was replaced with ColB1 (i), CaeA3 was replaced with ColA3 (ii), CaeA3 was replaced with ColA3 and additional ColG2 (iii), ColG2 was added (iv), and CaeA3 was replaced with CaeA3$_C$ColA3$_{A-PCP}$ (v). Synthetic **16** was used as a standard (vi).

ColA3) were observed in the biosynthetic pathway of collismycins (e.g., COL-A, Fig. 1)[14,24,25], the 2,2′-bipyridine antibiotics differing from CAEs with a sulfur-containing group at C5 of Ring **A**. Most likely, COLs share a similar assembly line for 2,2′-bipyridine formation in which, however, the dethiolation step can be skipped to retain the sulfur-containing group during assembly of the identical substrates picolinic acid, malonyl-CoA, L-cysteine, and L-leucine[16].

With great interest in the catalytic logic and the mechanism for sulfur fate determination during 2,2′-bipyridine formation, in this study, we reconstituted the assembly line for dethiolated antibiotics CAEs, validated the generality of 2,2′-bipyridine formation in the biosynthesis of thiolated antibiotics COLs and experimentally dissected the online process for 2,2′-bipyridine formation and differentiation. The findings reported here exemplify the complexity and variety of assembly line chemistry, which has long been an area of intense interest in the creation of structurally diverse nonribosomal peptides, polyketides, and their hybrid molecules.

## Results

**2,2′-Bipyridine assembly line functions with a *trans* flavoprotein.** We began by reconstituting the activities of the didomain NRPS CaeA1 (650-aa), the PKS-NRPS hybrid CaeA2 (2484-aa), and the single-module NRPS CaeA3 (1047-aa) in vitro (Fig. 2). These multifunctional enzymes were individually produced in an engineered *Escherichia coli* strain expressing Sfp[26,27], a phosphopantetheinyl (Ppant) transferase (PPTase) from *Bacillus subtilis*, leading to the in vivo conversion of their PCP and ACP domains from inactive *apo*-form into active *holo*-form by the thiolation of each active site L-serine residue with Ppant. The CaeA1, CaeA2, and CaeA3 proteins produced here were soluble (Supplementary Fig. 1), and believed to be active and able to sequentially incorporate the starter and extender units via thiolated PCP/ACP *S*-(amino)acylation for chain elongation (Supplementary Fig. 2a). However, no offline products were observed during the incubation with the substrates picolinic acid, malonyl-CoA, L-cysteine, and L-leucine in the presence of adenosine triphosphate (ATP, necessary for A domain activity) (Fig. 2a, i). The CAE-related assembly line likely functions with certain partner(s) that act(s) in trans.

Analyzing the *cae* cluster revealed two genes, *caeB1 and caeA4* (Fig. 1a), which encode a flavin-dependent protein and a type II thioesterase, respectively. Both genes are immediately downstream of *caeA2* and *caeA3* and are likely co-transcribed with

members after L-leucinyl removal, carboxyl reduction, and transamination[19–23]. If **1** is the direct product of CaeA1, CaeA2, and CaeA3, 2,2′-bipyridine formation could occur in this assembly line through a dethiolation-coupled heterocyclization process that involves NRPS-mediated C–C bond formation.

Interestingly, homologs of the above modular synth(et)ases CaeA1, CaeA2, and CaeA3 (i.e., ColA1a and ColA1b, ColA2, and

these upstream assembly line-encoding genes. To determine whether CaeB1 and CaeA4 are functionally related, we expressed them in *E. coli* (Supplementary Fig. 1). Purified CaeB1, which appeared light yellow, was determined to bind oxidized flavin adenine dinucleotide (FAD) in a noncovalent manner based on absorbance spectrum analysis and protein heating followed by high-performance liquid chromatography with high-resolution mass spectrometry (HPLC-HR-MS) (Supplementary Fig. 3). Indeed, the combination of CaeB1 with CaeA1, CaeA2, and CaeA3 resulted in the CAE intermediate 2,2'-bipyridinyl-L-leucine (**1**), whose production, however, did not require CaeA4-associated, putative thioesterase activity (Fig. 2a). Consequently, the CAE NRPS-PKS assembly line requires the *trans* activity of flavoprotein CaeB1 in the assembly of picolinic acid, malonyl-CoA, L-cysteine, and L-leucine. In the reaction mixture containing tris(2-carboxyethyl)phosphine (TCEP) as a reducing agent, we observed tris(2-carboxyethyl)phosphine-sulfide (**2**, Supplementary Fig. 4a), which was indicative of $H_2S$ production. $H_2S$ was then trapped using a fluorescent probe (**19**)[28] in the TCEP-free reaction mixture (Supplementary Fig. 4b), confirming the involvement of dethiolation in the production of **1**.

**The *trans* flavoprotein oxidatively processes L-cysteinyl on PCP.** Phylogenetically, CaeB1 is close to flavin-dependent dehydrogenases, particularly, those acting on acyl thioester substrates (Supplementary Fig. 3d)[29,30]. This observation led to the hypothesis that CaeB1 oxidatively processes L-cysteinyl on PCP prior to its noncanonical incorporation into Ring A of CAEs. We thus focused on CaeA2 and analyzed its changes in PCP *S*-aminoacylation using a bottom–up proteomics strategy[31–33]. For method establishment, thiolated CaeA2 was subjected to proteolysis with various proteases, and resultant peptide mixtures were analyzed by nanoLC-MS/MS to map the predicted sequences containing the Ppant-modified L-serine residue. While treating CaeA2 with trypsin, chymotrypsin, or Glu-C alone failed to produce related MS-detectable sequences, extensive attempts revealed a weak MS signal for **SLGGDSIMGIQF₂₀₄₂VSR** (the relevant Ser residue is underlined), a 15-aa sequence generated from complete trypsin treatment and partial chymotrypsin digestion (Supplementary Fig. 5a). The predicted **SLGGDSIMGIQF** peptide, an 12-aa sequence (shortened by 3-aa at C-terminus) resulting from the complete digestion with trypsin and chymotrypsin, was not detected, indicating the necessity of retaining these four residues, particularly the Arg alkaline residue that facilitates peptide ionization. To robustly produce an MS-detectable variant of the Arg-containing, 15-aa sequence for MS detection, we engineered CaeA2 and eliminated the remaining chymotrypsin cleavage site in **SLGGDSIMGIQF₂₀₄₂VSR** by mutating F2042 to residues Leu, Ile, and Val based on residue conservation analysis (Supplementary Fig. 2b). The activity of each resulting variant, i.e., CaeA2^F2042L, CaeA2^F2042I, or CaeA2^F2042V, was then assayed by substituting for wild-type CaeA2 in the **1**-producing reaction mixture. In contrast to the mutations F2042I and F2042V, both of which caused a significant decrease in **1** production, F2042L had little effect on CaeA2 activity (Supplementary Fig. 2b). In addition, similar to F2042I, the mutation F2042L did not affect protein production in *E. coli* and/or protein stability (Supplementary Fig. 1b). As a result, the CaeA2^F2042L variant, which can be completely digested with trypsin and chymotrypsin to efficiently produce the sequence **SLGGDSIMGIQL₂₀₄₂VSR**, was chosen for further analyses by HR-MS/MS to confirm the PCP sequence identity and covalently tethered substrate/intermediate (Supplementary Fig. 5b).

PCP *S*-aminoacylation-caused mass changes in **SLGGDSIM-GIQL₂₀₄₂VSR** were then monitored (Fig. 3). The incubation of

thiolated CaeA2^F2042L with L-cysteine yielded L-cysteinyl-*S*-CaeA2^F2042L (**3**) (Supplementary Fig. 6a), consistent with the activities of the A and PCP domains of the CaeA2 NRPS module. The presence of L-cysteinyl in CaeA2^F2042L was confirmed by treatment with iodoacetamide (IAA, a thiol derivatization reagent), which yielded derivative **4** (with an MS signal ~30-fold stronger than that of **3**, Fig. 3 (ii, left) and Supplementary Fig. 6b). Incubation of the above mixture with CaeB1 significantly lowered the production of **3** by ~70% (Fig. 3, iii, left). Accordingly, treatment with IAA generated **5**, a derivative of (3-sulfhydryl)-pyruvoyl-*S*-CaeA2^F2042L (**6**) (Fig. 3 (iii, middle) and Supplementary Fig. 7a), indicating that CaeB1 acts on PCP for L-cysteinyl transformation. This conclusion was supported by the observation of a +3 Da derivative of **5** when L-[1,2,3-$^{13}C_3$,$^{15}N$]cysteine was used (Supplementary Fig. 7). Using L-[2,3,3-$D_3$]cysteine as the substrate resulted in deuterium-unlabeled **5** (Supplementary Fig. 7), consistent with the notion that CaeB1 catalyzes the α,β-dehydrogenation of L-cysteinyl and converts **3** to dehydrocysteinyl-*S*-CaeA2^F2042L (**7**), which appears to be unstable and readily undergoes epimerization followed by hydrolysis to produce **6**. Proton exchange can occur at the Cβ position of **6**, leading to a loss of deuterium.

The sequence alignment of CaeB1 with various flavin-dependent dehydrogenases revealed potential key catalytic residues including E372 at the active site and S168 related to FAD binding (Supplementary Fig. 3d). These residues were mutated, generating CaeB1^E372L and CaeB1^S168A. The former variant shares with the wild-type protein a light yellow color and an absorbance spectrum characteristic of FAD in oxidized form; in contrast, the latter variant is nearly colorless, indicating the loss of FAD-binding ability (Supplementary Fig. 3). CaeB1^E372L and CaeB1^S168A were individually used to replace wild-type CaeB1 in the **1**-producing reaction mixture. Both mutations abolished **1** production, and saturation with an excess of FAD compensated the mutation S168A only (Supplementary Fig. 3). These results ascertained that CaeB1 activity is FAD-dependent. Mechanistically, CaeB1 catalyzes α,β-dehydrogenation likely utilizing E372 as a base for L-cysteinyl α-deprotonation, followed by the transfer of a β-hydride equivalent to FAD to yield dehydrocysteinyl and FADH₂ (Supplementary Fig. 8).

**The atypical NRPS Ct domain recruits *trans* flavoprotein activity.** Notably, CaeB1 activity strictly depends on the Ct domain of CaeA2, a C-like domain that is shortened by ~1/3 and lacks the catalytic L-histidine residue (Supplementary Fig. 9). Deleting this domain completely abolished the production of **1** (Fig. 2b, i). In the incubation of truncated CaeA2^F2042LΔCt with L-cysteine, L-cysteinyl-*S*-CaeA2^F2042LΔCt (**8**) was observed; however, the α,β-dehydrogenation of L-cysteinyl failed to occur when CaeB1 was introduced (Fig. 3 (iv, left and middle) and Supplementary Fig. 10). We thus propose that the Ct domain, which resembles the X domain of the NRPSs involved in glyco-peptide biosynthesis[34], is necessary for recruiting the activity of the flavoprotein CaeB1 that functions in trans. To validate this hypothesis, we expressed the *N*-terminally thioredoxin (Trx)-tagged Ct domain, the PCP domain, and the Ct-PCP didomain of the CaeA2 NRPS module in *E. coli* and measured their interactions with CaeB1 by isothermal titration calorimetry (ITC). Because CaeB1 is relatively unstable and tends to precipitate during the measurement, we prepared the variant of this flavoprotein by fusion with maltose-binding protein (MBP) at the *N*-terminus. ITC analysis revealed comparable $K_d$ values when titrating MBP-fused CaeB1 variant to the Trx-tagged Ct domain and the Ct-PCP didomain, respectively, as shown by 0.53 ± 0.12 μM and 0.42 ± 0.05 μM, indicating that a moderate interaction of

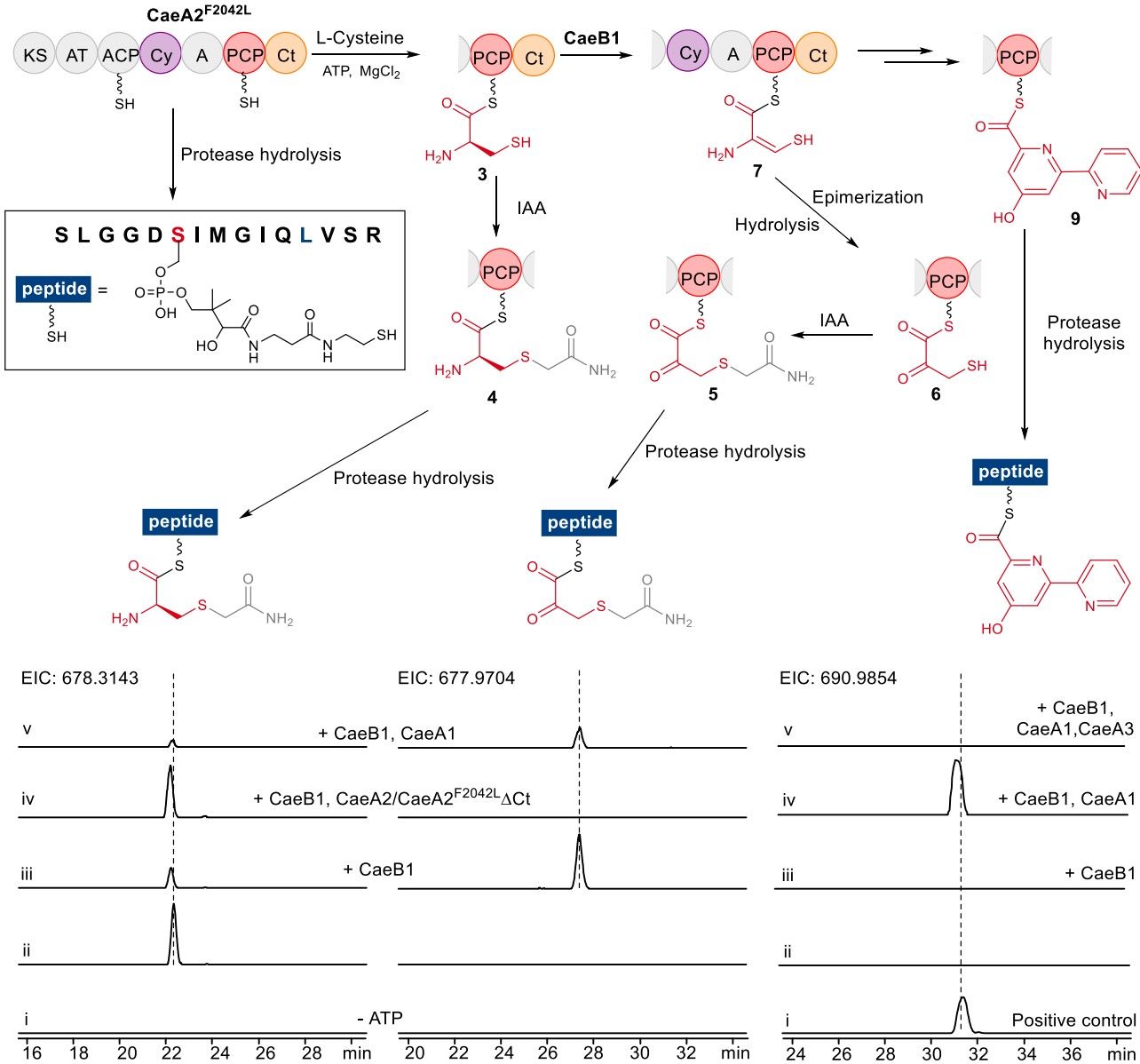

**Fig. 3 Examination of PCP-tethered intermediates during 2,2'-bipyridine formation (top) by nanoLC-MS/MS (below).** The sequence **SLGGDSIMGIQL₂₀₄₂VSR** (in the rectangle) arises from the complete digestion of CaeA2^F2042L with trypsin and chymotrypsin. For details of the HR-MS/ MS analyses, see Supplementary Figs. 5~7, 10, and 13. The two IAA-treated sequences (ESI $m/z$ [M + 3H]$^{3+}$ for the left, calcd. 678.3143; for the middle, calcd. 677.9704) come from L-cysteinyl-S-CaeA2^F2042L (**3**) and dehydrocysteinyl-S-CaeA2^F2042L (**7**), respectively, which were examined by incubating CaeA2^F2042L and L-cysteine in the absence (i, negative control) and presence (ii) of ATP, with CaeB1 (iii), CaeB1 and CaeA2^F2042LΔCt (replacing CaeA2^F2042L) (iv), or CaeB1, CaeA1, picolinic acid and malonyl-CoA (v). The right sequence (ESI $m/z$ [M + 3H]$^{3+}$, calcd. 690.9854) comes from 2,2'-bipyridinyl-S-CaeA2^F2042L (**9**). The positive control was prepared by incubating CaeA2^F2042L (with PCP in apo form) and 2,2'-bipyridinyl-S-CoA (i). **9** was examined by incubating CaeA2^F2042L and L-cysteine (ii), with CaeB1 (iii), CaeB1, CaeA1, picolinic acid, and malonyl-CoA (iv), or CaeB1, CaeA1, CaeA3, picolinic acid, malonyl-CoA and L-leucine (v). All examinations were performed at least in triplicate, and each had at least two parallel samples.

CaeB1 occurred with the Ct domain instead of the PCP domain (Fig. 4 and Supplementary Fig. 12). Consistent with these results, interactions were not observed by titrating the MBP-fused CaeB1 variant to control proteins, e.g., CaeA2ΔCt and the PCP domain alone (Supplementary Figs. 11 and 12). Consequently, the flavo-protein CaeB1 likely functions in trans through the interaction with the Ct domain of CaeA2, leaving the Cy domain in the atypical NRPS module as the catalyst for dehydrocysteinyl extension and subsequent heterocyclization.

**2,2'-Bipyridine formation proceeds in the atypical NRPS module.** We then examined the elongated intermediate

covalently tethered to the PCP domain of CaeA2 during **1** for-mation. The incubation of CaeA1, CaeA2^F2042L, and CaeB1 with picolinic acid, malonyl-CoA, and L-cysteine led to the accumu-lation of 2,2'-bipyridinyl-S-CaeA2^F2042L (**9**). In contrast, **9** was not observed in the presence of CaeA3 during **1** production (Fig. 3 (iv and v, right) and Supplementary Fig. 13a), supporting that 2,2'-bipyridinyl is an intermediate formed in the atypical NRPS module of CaeA2 before L-leucine extension. The atypical NRPS Cy domain likely catalyzes the formation of 2,2'-bipyr-idinyl, because mutating the catalytic residue D1165 in its con-served motif DxxxxD(1165)xxS to L-alanine completely abolished the production of **1** (Fig. 2b, ii). H₂S was observed during **9**

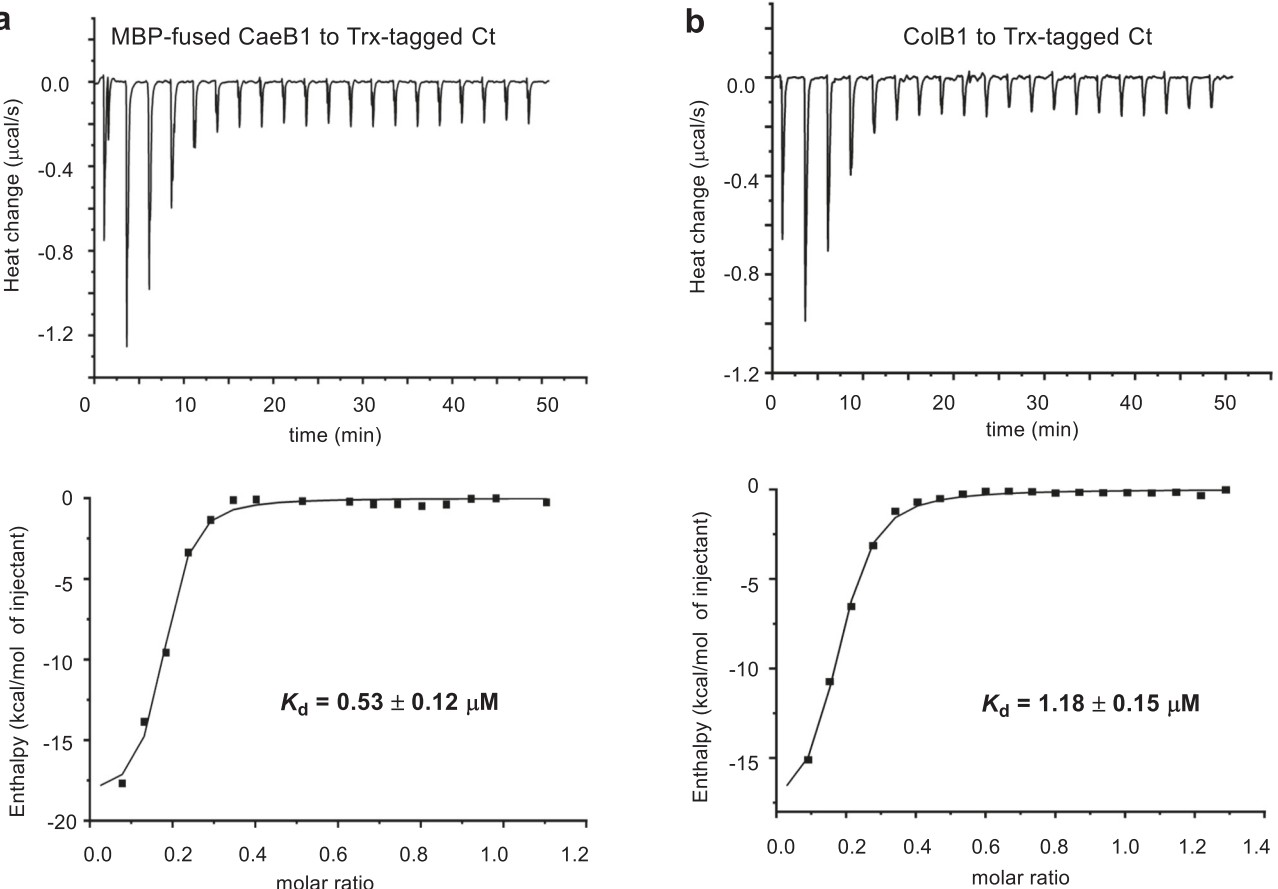

**Fig. 4 Measurement of the interactions of the Ct domain of CaeA2 with related flavoproteins by ITC.** Raw data were shown on top, and the integrated curves containing experimental points and the best fitting line obtained from the single binding site model were shown on the bottom. This measurement was conducted in triplicate. For negative controls, see Supplementary Fig. 11. **a** Titrating MBP-fused CaeB1 to Trx-tagged Ct. **b** Titrating ColB1 to Trx-tagged Ct.

formation (Supplementary Fig. 4a), indicating that dethiolation occurred. The production of 2,2'-bipyridinyl in the atypical NRPS module of CaeA2 was further confirmed by HR-MS/MS analysis of the positive control of 2,2'-bipyridinyl-$S$-CaeA2$^{F2042L}$, which was generated by incubating CaeA2$^{F2042L}$ with synthetic 2,2'-bipyridinyl-$S$-CoA in the presence of Sfp (Fig. 3 (i, right) and Supplementary Fig. 13b). In addition, the combination of CaeA3 with truncated 2,2'-bipyridinyl-$S$-PCP$_{CaeA2}$ (**10**, produced by thiolating the apo-form PCP$_{CaeA2}$ domain with Sfp in the presence of 2,2'-bipyridinyl-$S$-CoA) and L-leucine produced **1** (Supplementary Fig. 14), validating that this single-module NRPS accepts dethiolated 2,2'-bipyridinyl for L-leucine extension. Notably, 2,2'-bipyridinyl loses its chirality at Cα. CaeA3 possesses a C domain falling into the $^DC_L$ clade (Supplementary Fig. 15)[35], members of which typically catalyze the connection of an L-amino acid monomer to a growing peptidyl intermediate ending with a D-amino acid residue. The precedents of elongating Cα—achiral intermediate by a $^DC_L$ domain do exist, as exemplified in the biosynthesis of nocardicins[36], where a Cα—achiral intermediate is condensed with an L-amino acid for β-lactam formation.

**Timing of the dethiolation step in the formation of the CAE 2,2'-bipyridine core.** Focusing on the timing of the dethiolation step, we attempted to dissect the specific 2,2'-bipyridine-forming process in the CAE assembly line (Fig. 5). Omitting CaeA1 for picolinic acid incorporation from the **1** or **9**-forming reaction mixture prevented H$_2$S production (Supplementary Fig. 4a),

inconsistent with *route a* in which condensation follows the dethiolation of dehydrocysteinyl to 2-amino-allyl-$S$-CaeA2 (**11**). In addition, derivatives of the possible intermediate **11** (e.g., pyruvoyl-$S$-CaeA2, which might arise from epimerization and hydrolysis in a manner similar to the conversion of **7** to **6**) in this route were not observed based on careful HR-MS/MS analysis. Removing the substrate picolinic acid from the **1** or **9**-forming reaction mixtures led to similar results, supporting the hypothesis that the condensation with dehydrocysteinyl precedes its dethiolation. The Cy domain of CaeA2 shares high sequence homology to the counterparts known for thiazoline formation (Supplementary Figs. 9 and 16), in which related Cy domains display a dual activity for L-cysteine or L-serine extension by forming an amide bond first and then cyclization via the nucleophilic addition of the –SH or –OH side chain of the newly incorporated residue onto the preceding carbonyl group to yield a five-membered thiazoline or azoline ring after H$_2$O elimination[3,5]. However, the Cy domain of CaeA2 appears to be functionally distinct because it does not utilize unprocessed L-cysteinyl for condensation (Figs. 2 and 3). This domain likely follows CaeB1 activity and condenses the resulting reactive dehydrocysteinyl unit with the upstream intermediate for C–C bond formation through Cβ-nucleophilic substitution, yielding picolinyl-acetyl-dehydrocysteinyl-$S$-CaeA2 (**12**, Fig. 5) before heterocyclization to form the six-membered pyridine ring. We then utilized L-[2,3,3-D$_3$]cysteine in the **1**-forming reaction mixture. This assay led to the production of the +1 Da derivative of **1** (Supplementary Fig. 17). In *route b*, in which cyclization

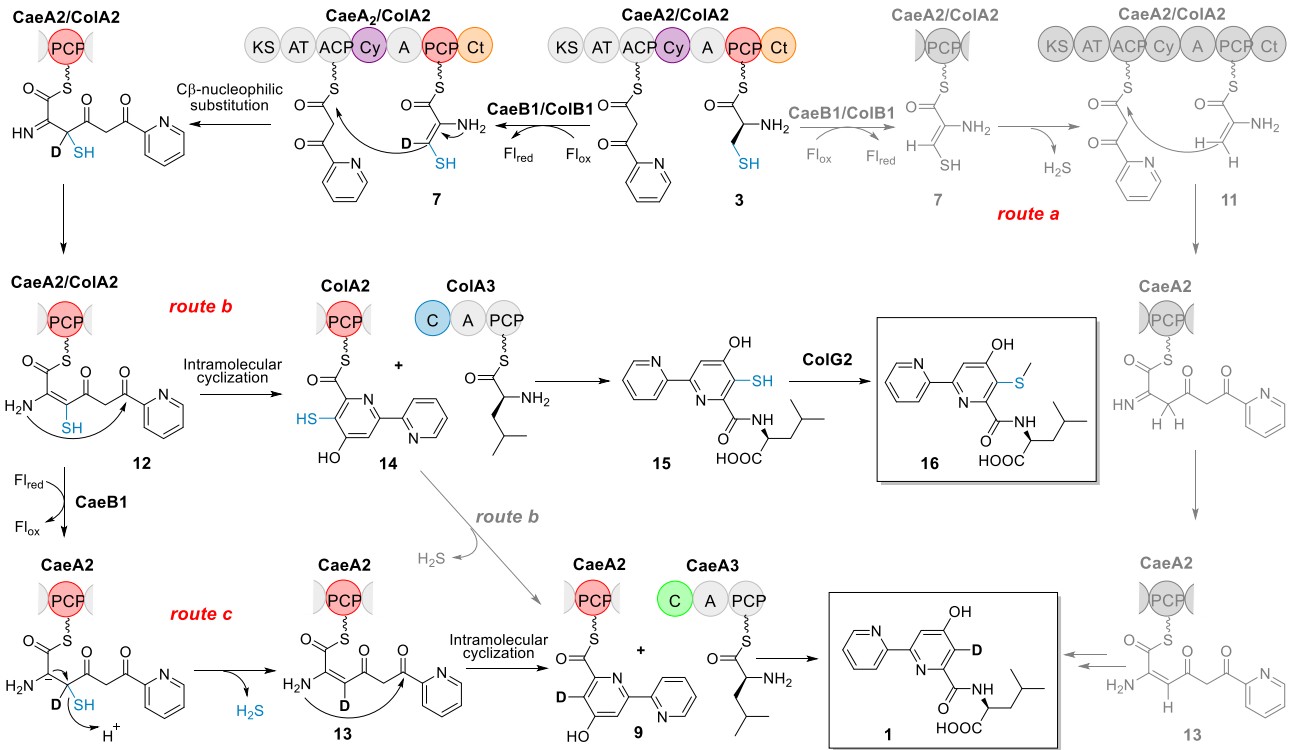

**Fig. 5 Proposed mechanisms for 2,2′-bipyridine formation.** The PCP domain of CaeA2 is labeled in red. The C domains of CaeA3 and ColA3 are highlighted in green and blue, respectively. In contrast to favored *routes b* and *c*, unfavored *route a* is shown by gray color. Fl$_{ox}$ oxidized flavin, Fl$_{red}$ two-electron reduced flavin.

precedes dethiolation, reductive dethiolation could be less likely to occur due to the more stable and less reactive aromatic pyridine ring system. Instead, *route c* is favored based on this observation because the reductive dethiolation of **12** to picolinyl-acetyl-dehydroalanyl-*S*-CaeA2 (**13**) mediated by CaeB1 reintroduces the previously abstracted deuterium at Cβ and thus facilitates cyclization to afford 2,2′-bipyridinyl (Fig. 5). For the dethiolation step, reduced FADH$_2$ might serve as a hydride supplier and undergo oxidation to produce FAD for recycling, consistent with a net non-redox process (Supplementary Fig. 8a).

**2,2′-Bipyridine assembly line provides both dethiolated and thiolated intermediates.** Intriguingly, although *route b* is not favored in CAE biosynthesis, skipping the dethiolation step allows the condensation of 5-sulfhydryl-2,2′-bipyridinyl-*S*-CaeA2 (**14**) with L-leucine to produce 5-sulfhydryl-2,2′-bipyridinyl-L-leucine (**15**), the proposed offline intermediate in the biosynthesis of COLs (Fig. 5). Distinct from CAEs, COLs are 2,2′-bipyridine antibiotics possessing a sulfur-containing group (Fig. 1). In addition to ColA1a and ColA1b, ColA2, and ColA3, which share high sequence homology with the CAE-related modular synth(et)ases CaeA1, CaeA2, and CaeA3, respectively, the biosynthetic pathway of COLs involves ColB1, the homolog of *trans*-acting flavoprotein CaeB1 that also noncovalently binds a FAD cofactor (Supplementary Fig. 3). Thus, the biosynthesis of COLs is assumed to share a similar *trans*-acting flavoprotein-dependent NRPS-PKS assembly line for the formation of the different sulfhydryl-2,2′-bipyridine core[14,25]. To determine the similarity in 2,2′-bipyridine formation, we attempted to replace the CAE enzymes with their counterparts from the COL pathway in the **1**-producing reaction mixture. These counterparts did not include those for picolinyl priming (i.e., CaeA1 or ColA1a/ColA1b) and subsequent two-carbon elongation (i.e., the PKS module of

CaeA2 or ColA2), which are believed to be identical in the biosynthesis of CAEs or COLs. The *trans* partners ColB1 and CaeB1 were confirmed to be interchangeable in the production of **1** (Fig. 2c, i), supporting that they both share the flavin-dependent activity for oxidatively processing L-cysteinyl to dehydrocysteinyl on PCP. Consistent with this finding, ITC analysis showed that ColB1 can interact with the Ct domain of the CaeA2, with a moderate $K_d$ value of $1.18 \pm 0.15\,\mu M$ (Fig. 4). Interestingly, changing the NRPS CaeA3 to ColA3 completely abolished **1** production (Fig. 2c, ii), indicating that ColA3 differs from CaeA3, even though they both mediate L-leucine extension.

In vitro testing of the exchangeability between the PKS-NRPS hybrid proteins CaeA2 and ColA2 failed because ColA2 was highly resistant to various methods for soluble protein preparation. We thus turned to in vivo biochemical assays and examined this exchangeability in the COL-producing strain *Streptomyces roseosporus* (Fig. 6). A chimeric gene that encodes the variant harboring the PKS module of ColA2 and the NRPS module of CaeA2, *colA2$_{PKS}$-caeA2$_{NRPS}$*, was constructed and introduced into the previously developed, △*colA2* mutant *S. roseosporus* strain that is incapable of producing COLs[14]. Remarkably, this complementation restored the production of COLs, supporting that the atypical NRPS modules of CaeA2 and ColA2 are functionally identical in terms of L-cysteinyl incorporation, *trans* CaeB1/ColB1 activity recruiting, condensation and heterocyclization during COL biosynthesis. These findings demonstrate the generality of both COLs and CAEs in 2,2′-bipyridine formation and particularly suggest that dethiolation can be skipped in the atypical NRPS module of CaeA2 to produce 5-sulfhydryl-2,2′-bipyridinyl. In addition, HPLC-HR-MS analysis revealed a trace of 2,2′-bipyridine carboxylate (**20**) in the wild-type *S. roseosporus* strain (Supplementary Fig. 18), indicating that both thiolated and dethiolated 2,2′-bipyridine intermediates can be formed in the COL assembly line. While the correct sulfhydryl-2,2′-bipyridinyl

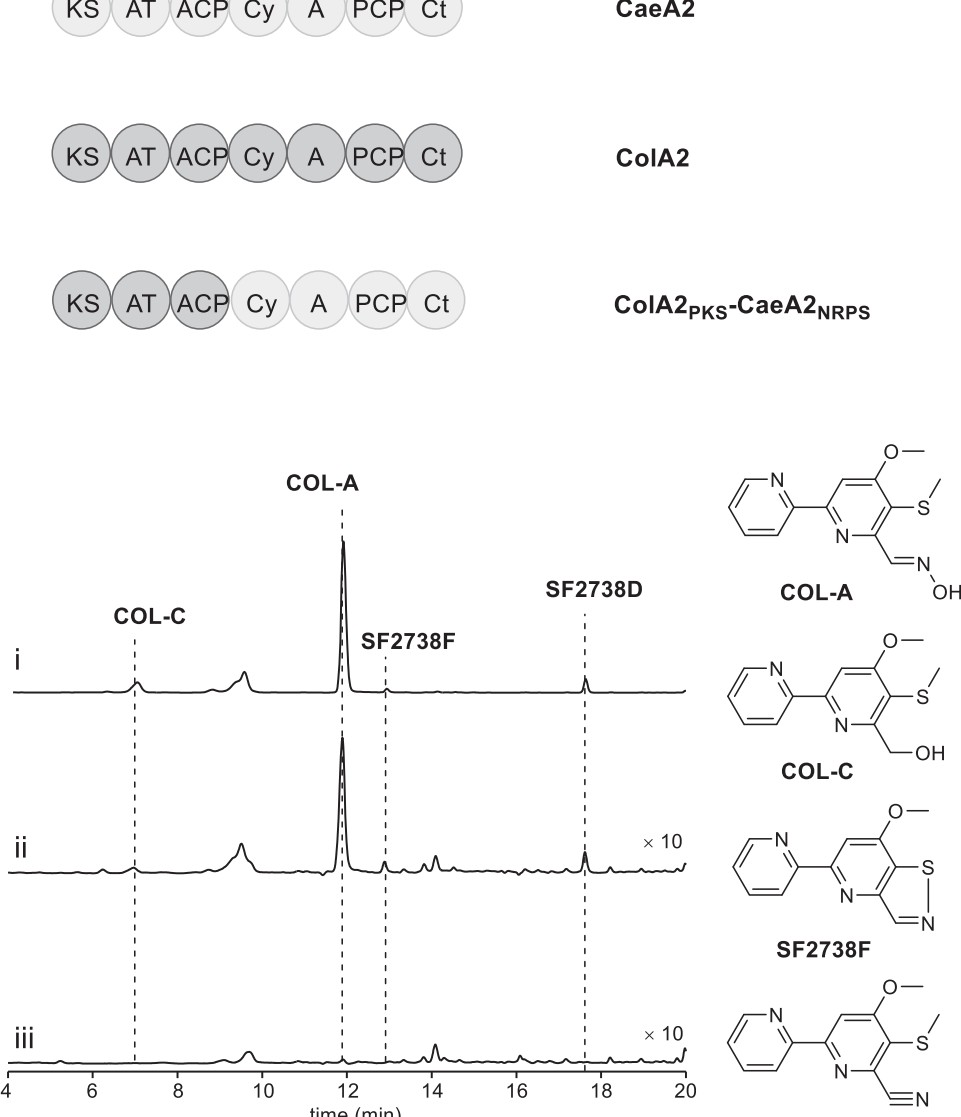

**Fig. 6 Domain organization of the NRPSs CaeA2, ColA2, and ColA2$_{PKS}$-CaeA2$_{NRPS}$ (Top) and HPLC analysis of COL-related products in *S. roseosporus* strains (below).** i, the wild-type COL-producing strain (positive control); ii, the ΔcolA2 mutant in which a chimeric gene *colA2$_{PKS}$-caeA2$_{NRPS}$* is expressed in trans; and iii, the Δ*colA2* mutant (negative control).

intermediate can be further condensed with L-leucine towards the biosynthesis of COLs, the incorrect 2,2'-bipyridinyl intermediate cannot and thus is subjected to hydrolysis from the assembly line. Most likely, the specificity of subsequent L-leucine extension has a role in pathway differentiating.

**L-Leucine extension specializes 2,2'-bipyridine differentiation.** Unlike 2,2'-bipyridinyl, sulfhydryl-2,2'-bipyridinyl formed in the assembly line is highly reactive and proved to be difficult for analysis by the aforementioned HR-MS/MS approach. To provide in vitro evidence to support the occurrence of this covalently tethered thiol intermediate, we attempted to examine the downstream offline intermediates by incubating CaeA1, CaeA2, and CaeB1 with ColA3 in the presence of the necessary substrates and cofactors. Treating the reaction mixture with IAA revealed **17**, the derivative of the expected product 5-sulfhydryl-2,2'-bipyridinyl-L-leucine (**15**) (Supplementary Fig. 19). To further validate the production of **15**, we added ColG2, a COL pathway-specific

methyltransferase[14,25], into this reaction mixture. After L-leucine extension, ColG2 is believed to catalyze the S-methylation of the highly reactive thiol of intermediate **15** to produce the less reactive, previously characterized intermediate 5-methylmercapto-2,2'-bipyridinyl-L-leucine (**16**). Indeed, **16** was observed as the sole product (Fig. 2c, iii). In contrast, the combination of ColG2 with the complete CAE assembly line (i.e., CaeA1, CaeA2, and CaeA3 as well as CaeB1) failed to produce **16** and had little effect on the production of **1** (Fig. 2c, iv), indicating that **15** cannot be generated in the presence of CaeA3. During the formation of the COL sulfhydryl-2,2'-bipyridine intermediate in the assembly line, where a net oxidation is necessary (Supplementary Fig. 8b), cycling reduced FADH$_2$ back to FAD could differ from that occurring in the formation of the CAE 2,2'-bipyridine intermediate and needs an exogenous oxidant, e.g., O$_2$ for in vitro assays. To validate this difference, we determined the O$_2$-dependence of CAE or COL biosynthesis by examining the production of **1** or **16** in vitro under anaerobic conditions (Supplementary Fig. 20). As expected, in the absence of O$_2$, CAE

intermediate **1** was clearly produced, whereas COL intermediate **16** was not observed.

We then narrowed the specificity of the upstream intermediate selection by CaeA3 and ColA3 to their C domains by domain swapping. Extensive attempts pointed to active CaeA3$_C$ColA3$_{A-PCP}$, the chimeric protein composed of the C domain of CaeA3, and the A-PCP didomain from ColA3. This protein was confirmed to be functionally identical to CaeA3 in the production of **1**, albeit with a lower yield (~70% decrease) (Fig. 2a, xi). It cannot be used to replace ColA3 because **16** was not observed in the presence of ColG2. Although the N-terminal docking regions (DRs) of CaeA3 and ColA3 differ from each other in size, they are conserved and share a β-sheet based on advance structure prediction (Supplementary Fig. 21). To examine whether their intermolecular interactions with the upstream NRPS modules play a role in differentiating the production of CAEs and COLs, we further prepared CaeA3-N' and ColA3-N', the two variants of CaeA3 and ColA3 with an exchanged N-terminal DR (Supplementary Fig. 1b). These variants were used individually to replace archetypal CaeA3 in the above reaction mixture containing the enzymes CaeA1, CaeA2, CaeB1, and ColG2 as well as necessary substrates and cofactors. Consequently, DR swapping did not change product profiles. CaeA3-N' and ColA3-N' are functionally equivalent to CaeA3 and ColA3, respectively, as only **1** or **16** was observed (Supplementary Fig. 21). Clearly, the CAE and COL assembly lines are identical for 2,2'-bipyridine formation and can simultaneously provide the dethiolated and thiolated intermediates (**9** and **14**), although their preferences could be different. The only exception is the C domain-conferred specificity in L-leucine extension, which specializes in the pathway to the production of either CAEs or COLs.

## Discussion

In this study, we demonstrate an unusual paradigm for 2,2'-bipyridine formation, which features an NRPS/PKS hybrid assembly line for sequential incorporation of the substrates picolinic acid, malonyl-CoA, L-cysteine, and L-leucine with the association of a flavoprotein partner functioning in trans (Fig. 5). This assembly line was first reconstituted in vitro for the biosynthesis of dethiolated 2,2'-bipyridine antibiotics CAEs. Then, the generality of 2,2'-bipyridine formation was validated in the biosynthesis of distinct sulfhydryl-2,2'-bipyridine antibiotics COLs, where the dethiolation step is skipped.

In the 2,2'-bipyridine assembly line, the atypical NRPS module Cy-A-PCP-Ct plays a central role (Fig. 5). Following the A domain activity of this module for L-cysteine incorporation, the Ct domain recruits the activity of a flavoprotein that functions in trans for oxidatively processing L-cysteinyl on PCP. The Cy domain, which differs from the counterparts well-known for thiazoline formation, then catalyzes C–C bond formation through unusual Cβ-nucleophilic substitution and condenses the resulting reactive dehydrocysteinyl extender unit with the upstream picolinyl-acetyl intermediate, yielding a linear picolinyl-acetyl-dehydrocysteinyl intermediate (**12**). In the COL pathway, where this intermediate can be immediately subjected to intramolecular cyclization to form a sulfhydryl-2,2'-bipyridine core, reduced FADH$_2$ (i.e., Fl$_{red}$) needs to be processed to the oxidized form (i.e., Fl$_{ox}$) by an oxidant for continuous production, consistent with the fact that sulfhydryl-2,2'-bipyridine intermediate **16** cannot be produced in vitro in the absence of O$_2$. In the CAE pathway, the flavin redox cycle is likely coupled to the dethiolation step via hydride transfer from Fl$_{red}$ to intermediate **12** and thus can be independent of an exogenous oxidant for Fl$_{ox}$ regeneration. This process facilitates H$_2$S elimination and the production of a distinct picolinyl-acetyl-dehydroalanyl

intermediate (**13**), which then undergoes heterocyclization to provide a dethiolated 2,2'-bipyridine core. In the assembly line, subsequent L-leucine extension, which does not contribute any atoms in the final molecules, determines the sulfur fate and thus differentiates the biosynthesis of CAEs and COLs (Fig. 5). Specifically, the C domain of the NRPS module confers this selectivity for L-leucine extension and advances one of the two intermediates down a path to the 2,2'-bipyridine antibiotics with or without sulfur decoration.

Although C domain selectivity for L-leucine extension has a critical gatekeeping role in sulfur fate determination, the fact that the CAE and COL assembly lines can provide two types of 2,2'-bipyridine intermediates generates interest in how they work effectively and economically during the specific biosynthesis of CAEs or COLs. We did not observe significant amounts of the respective shunt products in the native producers, suggesting that the incorrect 2,2'-bipyridine intermediate, either dethiolated or thiolated, is not accumulated in each assembly line. One possibility is that trans flavoprotein activity-dependent dethiolation is a reversible reaction between linear intermediates **12** and **13**. These reactive intermediates could largely stay in the assembly line, to avoid waste by minimizing the production of 2,2'-bipyridines **9** and **14** via heterocyclization, which appears to be not a reversible reaction, before selection by L-leucine extension (Fig. 5). Accumulating the incorrect 2,2'-bipyridine intermediate could stall the biosynthesis of the correct 2,2'-bipyridine in the assembly line, where its off-loading might lead to the formation of the shunt product. With this manner, the pressure arising from C domain-conferred gatekeeping role through L-leucine extension, the step following heterocyclization for pathway furcation, can be alleviated. In addition, there might be biases in sulfur fate determination, as exemplified in the assembly line of CAEs, where dethiolation appears to be much more favored to occur than being skipped. Despite extensive attempts including IAA derivatization, we did not observe the sulfhydryl-2,2'-bipyridine intermediate in the atypical NRPS module of CaeA2, where this intermediate was proved to be available for ColA3-catalyzed L-leucine extension towards the biosynthesis of COLs, indicating its low yield. In the future work, characterizing how the Cy domain functionally collaborates with the trans-acting flavoprotein is likely necessary for determining the dethiolation step, where it is optional and relies on additional trans-acting flavoprotein activity-dependent activity for reduction between the Cy domain-catalyzed condensation with dehydrocysteinyl and the selective heterocyclization/2,2'-bipyridine formation.

In conclusion, we uncover a noncanonical trans-acting flavoprotein-dependent paradigm in assembly line chemistry, where cyclization reactions are usually difficult to be approached and thus remain poorly understood compared with those occurring in the offline stage[3,4]. Our studies advance the understanding of the versatile functions of C domains in NRPS catalysis[37]. These domains include the gatekeeping C domain for sulfur fate determination, the Ct domain for trans-acting flavoprotein activity recruiting, and, particularly, the unusual Cy domain for C–C bond formation and heterocyclization. The findings reported here further our appreciation of assembly line enzymology and will facilitate the design, development, and utilization of enzymatic molecular machinery to address synthetic challenges arising from structurally related complex polypeptides, polyketides, and their hybrids.

## Methods

**General**. General materials and methods used in this study are summarized in Supplementary methods. The bacterial strains, plasmids, and primers used in this study are summarized in Supplementary Tables 1–3, respectively.

**DNA manipulation and sequencing**. DNA isolation and manipulation in *E. coli* or actinobacteria were carried out according to standard methods. PCR amplifications were carried out on an Applied Biosystems Veriti Thermal Cycler (Thermo Fisher Scientific Inc., USA) using either Taq DNA polymerase (Vazyme Biotech Co. Ltd, China) for routine verification or PrimeSTAR HS DNA polymerase (Takara Biotechnology Co., Ltd. Japan) for high fidelity amplification. Primer synthesis was performed at Shanghai Sangon Biotech Co. Ltd. (China). DNA sequencing was performed at Shanghai Majorbio Biotech Co. Ltd. (China).

**Sequence analysis**. Open reading frames were identified using the FramePlot 4.0beta program (http://nocardia.nih.go.jp/fp4/). The deduced proteins were compared with other known proteins in the databases using available BLAST methods (http://blast.ncbi.nlm.nih.gov/Blast.cgi). Amino acid sequence alignments were performed using Vector NT1 and ESPript 3.0 (http://espript.ibcp.fr/ESPript/ESPript/).

**Protein expression and purification**. The recombinant proteins CaeA1, PCP$_{CaeA2}$, CaeB1, ColB1, ColG2, CaeA3, and ColA3, as well as the chimeric NRPS proteins CaeA3$_C$ColA3$_{A-PCP}$, CaeA3-N' and ColA3-N' that arise from domain swapping, were produced in a form tagged by 6×His at N-terminus or by both MBP and 6×His at N-terminus, whereas CaeA2 and its variants (e.g., CaeA2$^{F2042L}$, CaeA2$^{F2042I}$, CaeA2$^{F2042V}$, Ct $_{CaeA2}$, CaeA2$^{-\Delta Ct}$, CaeA2$^{F2042L}\Delta$Ct, and PCP-Ct$_{CaeA2}$) were expressed in a form tagged by 8×His at C-terminus, Trx and 6×His at N-terminus or Sumo and 6×His at N-terminus. For details, please see Supplementary methods.

**Site-specific mutation of CaeA2 and its truncated proteins**. Rolling cycle PCR amplification (using the primers listed in Supplementary Table 3) followed by subsequent DpnI digestion was conducted according to the standard procedure of the Mut Express II Fast Mutagenesis Kit (Vazyme Biotech Co. Ltd, China). The yielded recombinant plasmids were listed in Supplementary Table 2. Each mutation was confirmed by Sanger sequencing. CaeA2$^{F2042L}$, CaeA2$^{F2042L}\Delta$Ct, CaeA2$^{F2042I}$, and CaeA2$^{F2042V}$ were expressed in *E.coli* BAP1. All the mutant proteins were purified to homogeneity and then concentrated according to the procedures for the native proteins described above.

**Determination of the flavin cofactor**. Each protein (CaeB1 or ColB1) solution at the concentration of 1 mg/ml was incubated at 100°C for 5 min for denaturation and then subjected to HPLC-DAD analysis on an Agilent Zorbax column (SB-C18, 5 μm, 4.6 × 250 mm, Agilent Technologies Inc., USA) by gradient elution of solvent A (H$_2$O containing 20 mM ammonium acetate) and solvent B (CH$_3$CN) at a flow rate of 1 mL/min over a 35-min period as follows: $T = 0$ min, 5% B; $T = 2$ min, 5% B; $T = 20$ min, 90% B; $T = 25$ min, 90% B; $T = 30$ min, 5% B, and $T = 35$, 5% B ($\lambda$ at 448 nm), using standard FAD as control. The supernatant of CaeB1 or ColB1 was subjected to ESI-HR-MS analysis to confirmed the identity of FAD (ESI $m/z$ [M + H]$^+$, calcd. 786.1644; found 786.1593).

**Reconstitution of the 2,2′-bipyridine assembly line in vitro**. The initial reaction was conducted at 30°C for 1 h in a 100 μL reaction mixture containing 50 mM Tris-HCl (pH 7.5), 1 mM TCEP, 10 mM MgCl$_2$, 1 mM picolinic acid, 1 mM malonyl-*S*-CoA, 100 μM L-cysteine, 1 mM L-leucine, 10 μM CaeA1, 1 μM CaeA2, 1 μM CaeA3, and 4 mM ATP. To determine the necessary *trans* partner, CaeB1, CaeA4, or both of them were added into the above mixture, respectively, with the final concentration of 1 μM for each protein. Each reaction was quenched with 100 μL of CH$_3$CN after incubation. To examine the production of the dethiolated 2,2′-bipyridine intermediate **1**, reaction mixtures were subjected to HPLC analysis on an Agilent Zorbax column (SB-C18, 5 μm, 4.6 × 250 mm, Agilent Technologies Inc., USA) using a DAD detector, by gradient elution of solvent A (H$_2$O containing 0.1% TFA) and solvent B (CH$_3$CN containing 0.1% TFA) at a flow rate of 1 mL/min over a 35-min period as follows: $T = 0$ min, 5% B; $T = 2$ min, 5% B; $T = 20$ min, 90% B; $T = 25$ min, 90% B; $T = 30$ min, 5% B, and $T = 35$, 5% B ($\lambda$ at 315 nm). For HPLC-ESI-MS analysis, TFA was replaced by 0.1% formic acid.

For H$_2$S examination during **1** production, the reaction was conducted at 30°C for 1 h in the 100 μL, TCEP-involving reaction mixture that contains 50 mM Tris-HCl (pH 7.5), 1 mM TCEP, 10 mM MgCl$_2$, 1 mM picolinic acid, 1 mM malonyl-*S*-CoA, 100 μM L-cysteine, 1 mM L-leucine, 10 μM CaeA1, 1 μM CaeA2, 1 μM CaeA3, 1 μM CaeB1, and 4 mM ATP. Then, the TCEP derivative **2** was analyzed by HPLC-ESI-MS under conditions as described above. Alternatively, H$_2$S examination was conducted in the reaction mixture where TCEP was omitted, in the presence of **19** (1 μM), a dual-reactable fluorescent probe used for highly selective and sensitive detection of biological H$_2$S[5]. The reaction of **19** with Na$_2$S to yield **21** serves as the control reaction. **21** was examined by HPLC-FLD under conditions as described above with excitation at 370 nm and relative emission at 450 nm.

To examine whether the CAE assembly line provides the thiolated intermediate in COL biosynthesis, ColG2 (with the final concentration of 1 μM) and SAM (with the final concentration of 1 mM) were added into the 100 μL reaction mixture that contains 50 mM Tris-HCl (pH 7.5), 1 mM TCEP, 10 mM MgCl$_2$, 1 mM picolinic

acid, 1 mM malonyl-*S*-CoA, 100 μM L-cysteine, 1 mM L-leucine, 10 μM CaeA1, 1 μM CaeA2, 1 μM CaeB1, 1 μM CaeA3 (ColA3, CaeA3$_C$ColA3$_{A-PCP}$, CaeA3-N' or ColA3-N'), and 4 mM ATP. Reactions are conducted at 30°C for 1 h and then quenched with 100 μL of CH$_3$CN. The production of the thiolated 2,2′-bipyridine intermediate **16** was examined by HPLC or HPLC-ESI-MS under conditions as described above.

For sulfhydryl-2,2′-bipyridinyl-L-leucine (**15**) examination, the reaction was conducted at 30°C for 1 h in the 50 μL reaction mixture contains 50 mM Tris-HCl (pH 7.5), 1 mM TCEP, 10 mM MgCl$_2$, 1 mM picolinic acid, 1 mM malonyl-*S*-CoA, 100 μM L-cysteine, 1 mM L-leucine, 10 μM CaeA1, 1 μM CaeA2, 1 μM ColA3, 1 μM CaeB1, and 4 mM ATP. The reaction mixture was treated with 1% sodium dodecyl sulphate and 5 mM TCEP at 55°C for 30 min to release the free thiol of **15**, and then was incubated with 25 mM IAA at 30°C for 30 min. The production of **17** was examined by HPLC or HPLC-ESI-MS under conditions as described above.

To examine O$_2$ dependence under anaerobic conditions, gas exchange for O$_2$ elimination was conducted in an anaerobic glovebox overnight before the incubation of the related reaction mixture at 30°C for 1 h. All assays were performed at least in triplicate and each had at least two parallel samples.

**In vitro assays of PCP S-aminoacylation on CaeA2 by nanoLC-MS/MS**. The reaction mixtures were subjected to complete protease hydrolysis with trypsin and chymotrypsin to map **SLGGGDSIMGIQL$_{2042}$VSR** of CaeA2, the MS-detectable sequence that contains the Ppant-modified active site L-serine residue (underlined). The digestion mixtures were filtrated using Microcon YM-10 (MilliporeSigma, USA) by centrifugation and stored at −80°C before analysis. For nanoLC-MS/MS analysis, each sample was loaded on a trap column (75 μm i.d., 2 cm, C18, 5 μm, 100 Å, Thermo Fisher Scientific Inc., USA) for online desalting, and then was separated using a reversed-phase column (75 μm i.d., 10.2 cm, C18, 3 μm, 120 Å, New Objective Inc., USA) by gradient elution of solvent A (H$_2$O containing 0.1% formic acid) and solvent B (80% CH$_3$CN containing 0.1% formic acid) at a flow rate of 300 nL/min over a 1.5 h period as follows: $T = 0$ min, 18% B; $T = 45$ min, 45% B; $T = 50$ min, 100% B; and $T = 90$ min, 100% B. For MS analysis, the nano-ESI voltage and capillary temperature were set at 2.2 kV and 275°C, respectively. The MS data were acquired in data-dependent mode. Each full-scan MS ($m/z$ 350 − 2000, resolution of 60 k) was followed with 10 HCD MS/MS scans (normalized collision energy of 33, resolution of 15 k) for the most intense precursor ions. The maximum ion injection time for MS and MS/MS were 50 and 45 ms, and the auto gain control target for MS and MS/MS were $3 \times 10^6$ and $5 \times 10^4$, respectively. The dynamic-exclusion time was set as 40 s.

**In vitro preparation of 2,2′-bipyridinyl-S-CaeA2$^{F2042L}$ (9)**. To prepare **9** as a positive control in *S*-aminoacylation assays, the reaction was conducted at 30°C for 30 min in a 50 μL reaction mixture containing 50 mM Tris-HCl (pH 7.5), 1 mM TCEP, 10 mM MgCl$_2$, 100 μM 2,2′-bipyridinyl-*S*-CoA, 5 μM Sfp, and 10 μM CaeA2$^{F2042L}$ in apo-form. Protease digestion and subsequent nanoLC-MS/MS analyses were conducted using approaches as described above.

**In vitro preparation of 2,2′-bipyridinyl-S-PCP$_{CaeA2}$ (10)**. The reaction was conducted at 30°C for 1 h in an 80 μL reaction mixture containing 62.5 mM Tris-HCl (pH = 7.5), 1.25 mM TCEP, 12.5 mM MgCl$_2$, 1 mM 2,2′-bipyridinyl-*S*-CoA, 100 μM PCP$_{CaeA2}$ (derived from wild-type CaeA2) in apo-form, and 4 μM Sfp.

**Measurement of protein–protein interactions by ITC**. ITC was performed with MicroCal-ITC200 (Malvern) at 25°C. A 400 μl aliquot of 60 μM PCP-Ct$_{CaeA2}$, Ct, PCP$_{CaeA2}$, or CaeA2$\Delta$Ct was placed in the stirred cell, and 120 μl aliquot of 400 μM MBP-fused CaeB1 or ColB1 was prepared in the syringe. All the recombinant proteins were prepared by Ni-affinity chromatography followed by size-exclusion chromatography for purification, and then were exchanged in the 50 mM Tris-HCl (PH = 7.5) buffer containing 100 mM NaCl, 0.1 mM TCEP, and 5% glycerol. The titration was performed as follows: 1 μl of protein in the syringe over 0.8 s for the first injection, followed by 19 injections of 2 μl protein in the stirred cell at 120 s intervals. The heat of reaction per injection was determined by integration of the peak areas using the MicroCal-PEAQ-ITC software, which provides the best-fit values for the heat of binding (ΔH), the stoichiometry of binding (N), and the dissociation constant ($K_d$). The heats of dilution were determined by injecting flavoprotein alone into the buffer and were subtracted from the corresponding experiments before curve fitting. All assays were performed at least in triplicate and each had at least two parallel samples.

**Analysis of the functional exchangeability between CaeA2 and ColA2 in vivo**. A 4.4 kb DNA fragment amplified by PCR from the CAE-producing *A. cyanogriseus* strain using the primers cae-CO-for and cae-Ct-rev was cloned into pMD19-T, yielding pQL1053. After sequencing to confirm the fidelity, this PCR product was recovered by digestion with BglII-EcoRV and then utilized to replace the 4.4 kb BglII-EcoRV fragment of pQL1022[14], a pSET152 derivative previously constructed for *colA2* expression under the control of *PermE** (the constitutive promoter for expressing the erythromycin-resistance gene in *Saccharopolyspora*

*erythraea*). The resulting recombinant plasmid pQL1054, which carries the chimeric gene *col/caeA2* coding for the hybrid protein that harbors the PKS module from ColA2 and the NRPS module from CaeA2, was transferred by conjugation into the △*colA2 S. roseosporus* mutant strain[7], yielding the recombinant strain QL2006. The fermentation of QL2006 and the examination of CAE or COL production were conducted according the methods described previously[14].

**Chemical synthesis.** The precursor 4-hydroxy-2,2'-bipyridinyl-6-carboxylic acid hydrobromide (**20**) was synthesized according to the method described previously[23]. Then **20** further undergone thiolation and thioester exchange reaction to obtain 2,2'-bipyridinyl-S-CoA. For details, please see Supplementary methods.

**Reporting summary.** Further information on research design is available in the Nature Research Reporting Summary linked to this article.

## Data availability

The data underlying the findings of this study are available in this article and its Supplementary Information or are available from the corresponding authors upon reasonable request. Source data are provided with this paper.

## Code availability

To probe peptide-tethered intermediates, all MS/MS data of the derivatives from the sequence (**SLGGDSIMGIQL**$_{2042}$VSR) were extracted from raw MS data using script written by Python, which is available in github: https://github.com/billpb610/AlFinder/blob/master/AlFinderS2.py.

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

## Acknowledgements

We thank Dr. L. Yi (Beijing University of Chemical Technology) and professor Z. Xi (Nankai University) for providing the fluorescent probe in H$_2$S examination. This work was supported in part by grants from the National Key R&D Program of China (2019YFA0905400), NSFC (32030002 and 21750004), CAS (QYZDJ-SSW-SLH037 and XDB20020200), and K.C. Wong Education Foundation.

## Author contributions

B.P., Z.T., and S.G. provide in vitro biochemical data; B.P. and Z.W. conducted in vivo experiments; B.P., R.L., Z.T., and W.L. designed the experiments and analyzed the data; W.L. wrote the paper and directed the research.

## Competing interests

The authors declare no competing interests.
