## [Peer Review File · Nature Communications]

Reviewers' Comments:

Reviewer #1:

Remarks to the Author:

The manuscript by Pang et al. describes the investigation of the assembly lines for the biosynthesis of the related NRPS-PKS hybrids collismycin and caerulomycin. In the third, vastly improved version of this manuscript, the authors have meticulously addressed all concerns and I therefore support publication of this work. Below are a few minor points that should be considered prior to publication.

Minor points:

p. 4, line 62: Correct "malonyl-S-CoA" to "malonyl-CoA" (several times in the manuscript) as previously pointed out.

p.4 lines 69 & 75: "lacks precedent" and "unprecedented" is repetitive. I suggest to use "unprecedented" not more than once.

p. 6, line 109: "type II" with small letter

p. 13, first paragraph: "This assay led to the production of the + 1Da derivative of 1 (Supplementary Fig. 17), which excludes route b in which cyclization precedes dethiolation. As shown in this route, cyclization requires the epimerization of 12 to the enamine, which could eliminate the C β deuterium of the dehydrocysteinyI residue and result in unlabeled 1."

Notably, the production of the +1 derivative does not strictly rule out route b, as the deuterium would also be re-introduced during conversion of 14 into 9; however, in this case the reductive desulfuration would be less likely to occur due to the more stable and less reactive aromatic pyridine ring system and therefore route a appears more likely. Please adjust the text section accordingly. Moreover as the deuterium is re-introduced, please rephrase the next section. For example write: "Instead, route c is favored based on this observation because the reductive dethiolation of 12 to picolinyl-acetyl-dehydroalanyl-S-CaeA2 (13) mediated by CaeB1 reintroduces the previously abstracted deuterium at C β and thus facilitates cyclization to afford 2,2'-bipyridinyl (Fig. 5)."

Response to Decision Letter

We would like to thank the reviewers and the editorial staff for taking the time to evaluate the manuscript, which has certainly improved as a result of the feedback. Below, we provide a point-by-point response to each comment with the reviewer's text in black and our response in blue.

p. 4, line 62: Correct "malonyl-S-CoA" to "malonyl-CoA" (several times in the manuscript) as previously pointed out.

Done.

p. 4, lines 69 & 75: "lacks precedent" and "unprecedented" is repetitive. I suggest to use "unprecedented" not more than once.

We have deleted the word "unprecedented" as the reviewer suggested.

p. 6, line 109: "type II" with small letter

Done.

p. 13, first paragraph: "This assay led to the production of the + 1Da derivative of 1 (Supplementary Fig. 17), which excludes route b in which cyclization precedes dethiolation. As shown in this route, cyclization requires the epimerization of 12 to the enamine, which could eliminate the C β deuterium of the dehydrocysteinyll residue and result in unlabeled 1."

Notably, the production of the +1 derivative does not strictly rule out route b, as the deuterium would also be re-introduced during conversion of 14 into 9; however, in this case the reductive desulfuration would be less likely to occur due to the more stable and less reactive aromatic pyridine ring system and therefore route a appears more likely. Please adjust the text section accordingly. Moreover as the deuterium is re-introduced, please rephrase the next section. For example write: "Instead, route c is favored based on this observation because the reductive dethiolation of 12 to picolinyl-acetyl-dehydroalanyl-S-CaeA2 (13) mediated by CaeB1 reintroduces the previously abstracted deuterium at C β and thus facilitates cyclization to afford

2,2'-bipyridinyl (Fig. 5).”

We have revised related sentences as the reviewer suggested. Please see Page 12 in the revised manuscript.